# Influence of Diabetes Mellitus on Postoperative Complications After Total Knee Arthroplasty: A Systematic Review and Meta-Analysis

**DOI:** 10.3390/medicina60111757

**Published:** 2024-10-26

**Authors:** Seok Ho Hong, Seung Cheol Kwon, Jong Hwa Lee, Shinje Moon, Joong Il Kim

**Affiliations:** 1Department of Orthopaedic Surgery, Kangnam Sacred Heart Hospital, Hallym University College of Medicine, Seoul 07441, Republic of Korea; shhong@hallym.or.kr (S.H.H.); youthinl@naver.com (S.C.K.); jongl525@naver.com (J.H.L.); 2Department of Internal Medicine, Hanyang University Seoul Hospital, Hanyang University College of Medicine, Seoul 04763, Republic of Korea

**Keywords:** total knee arthroplasty, diabetes mellitus, postoperative complication, mortality, meta-analysis

## Abstract

*Background and Objectives*: Total knee arthroplasty (TKA) is an effective treatment option for severe knee osteoarthritis. Understanding the impact of diabetes mellitus (DM) on postoperative outcomes is crucial for improving patient satisfaction after TKA. This study aimed to investigate the influence of DM on postoperative complications and mortality after TKA. *Materials and Methods*: We conducted a systematic review and meta-analysis by searching relevant studies published before December 2023 in the PubMed, EMBASE, Cochrane Library, Medline, and Web of Science databases. The assessment included demographic data, comorbidities, and postoperative complications after primary TKA for both DM and non-DM patients. The odds ratio (OR) was used to represent the estimate of risk of a specific outcome. *Results*: Thirty-nine studies were finally included in this meta-analysis. Patients with DM had higher rates of periprosthetic joint infection (OR: 1.71, 95% confidence interval [CI]: 1.46–2.00, *p* < 0.01) and prosthesis revision (OR: 1.37, 95% CI: 1.23–1.52, *p* < 0.01). Moreover, patients with DM showed an elevated incidence of pneumonia (OR: 1.54, 95% CI: 1.15–2.07, *p* < 0.01), urinary tract infection (OR: 1.86, 95% CI: 1.07–3.26, *p* = 0.02), and sepsis (OR: 1.61, 95% CI: 1.46–1.78, *p* < 0.01). Additionally, the postoperative risk of cardiovascular (OR: 2.49, 95% CI: 1.50–4.17, *p* < 0.01) and cerebrovascular (OR: 2.38, 95% CI: 1.48–3.81, *p* < 0.01) events was notably higher in patients with DM. The presence of DM increased the risk of deep vein thrombosis (OR: 1.58, 95% CI: 1.22–2.04, *p* < 0.01), but did not lead to an increased risk of pulmonary embolism. Most importantly, DM was associated with a higher mortality rate within 30 days after TKA (OR: 1.27, 95% CI: 1.02–1.60, *p* = 0.03). *Conclusions*: Patients with DM exhibited a higher rate of postoperative complications after TKA, and DM was associated with a higher mortality rate within 30 days after TKA. It is crucial to educate patients about the perioperative risk and develop evidence-based guidelines to prevent complications after TKA.

## 1. Introduction

Knee osteoarthritis can lead to progressive pain and dysfunction, significantly affecting a patient’s quality of life [1]. Total knee arthroplasty (TKA) is the treatment of choice for patients with severe knee osteoarthritis [2]. However, as the age of patients undergoing TKA increases, a considerable number develop multiple comorbidities [3,4]. These comorbidities can affect postoperative complications and are closely related to functional outcomes and quality of life after TKA [5,6,7]. Therefore, understanding the impact of comorbidities on postoperative complications and their prevention is essential to improve patient satisfaction after TKA.

Diabetes mellitus (DM) is a significant risk factor for postoperative complications following TKA [8]. Numerous studies have shown that DM increases the risk of periprosthetic joint infection and revision TKA [9,10,11]. Additionally, several studies have demonstrated that DM can elevate the risk of other systemic complications, such as urinary tract infections (UTI), pneumonia, sepsis, deep vein thrombosis/pulmonary embolism (DVT/PE), cardiovascular disease/cerebrovascular accidents (CVD/CVA), and mortality [12,13,14,15,16]. Yang et al. [16] found that the incidence of DVT after TKA was significantly higher in patients with DM compared to those without. Similarly, Martínez-Huedo et al. [15] reported a significantly higher incidence of UTI in DM patients following TKA. However, the limited number of studies and small sample sizes make it difficult to draw definitive conclusions. Therefore, clarifying the impact of DM on these systemic complications holds great clinical significance.

Moreover, recent advancements in perioperative patient management, such as enhanced recovery after surgery (ERAS), have significantly changed postoperative complication rates [17,18]. However, meta-analyses that incorporate these recent studies are scarce. Therefore, we aimed to conduct a meta-analysis of studies investigating the impact of DM not only on periprosthetic joint infection and revision rates, but also on systemic infections, DVT/PE, CVD/CVA, and mortality among TKA patients. This study is believed to be beneficial in establishing a strategy to predict, prevent, and manage TKA complications in patients with DM. The hypothesis of this study is that patients with DM not only demonstrate a higher risk of periprosthetic joint infection and higher revision rates but also an increased incidence of systemic complications and a higher mortality rate.

## 2. Materials and Methods

### 2.1. Search Strategy

This meta-analysis was performed in accordance with the Preferred Reporting Items for Systematic Reviews and Meta-Analyses (PRISMA) guidelines (Appendix A). The study protocol was registered in a prospective registry of systematic reviews (CRD number: 42024543515). Two investigators refined the data extraction tables before extraction and then searched citation databases, including PubMed, EMBASE, and Cochrane Library, from inception until 31 December 2023. The search terms included combinations of the following: (‘Total knee replacement’ or ‘Total knee arthroplasty’ or ‘TKA’) and ‘Diabetes’ and (‘Infection’ or ‘Revision’ or ‘Fracture’ or ‘Mortality’ or ‘Complication’ or ‘Cardiovascular’ or ‘CVD’ or ‘Cerebrovascular CVA’ or ‘or ‘CVA” or ‘Pneumonia’ or ‘Embolism’ or ‘Thrombosis’ or ‘Sepsis’).

### 2.2. Study Selection and Data Extraction

Studies with the following characteristics were included: (1) population: patients who underwent TKA; (2) intervention: patients with diabetes; (3) comparators: patients without diabetes; (4) outcomes: all-cause mortality and complications of TKA; and (5) study design: case–control studies. We excluded studies with the following characteristics: (1) articles on animal studies or in vivo experiments; (2) articles that included only abstracts; (3) non-original articles, including expert opinions or reviews; and (4) studies with insufficient information on TKA complications. The following variables were independently extracted by two investigators using the same criteria: first author, publication year, country, number of study participants, number of diabetes cases, mean age, sex ratio, mortality, and complications.

### 2.3. Quality Assessment

The Risk of Bias Assessment Tool for Non-Randomized Studies (RoBANS) was used to assess the methodological quality of case–control studies. The RoBANS evaluated the following parameters: (1) selection of participants, (2) confounding variables, (3) measurement of intervention, (4) blinding for outcome assessment, (5) incomplete outcome data, and (6) selective outcome reporting. These parameters were independently assessed by two reviewers. Any discrepancies were resolved by discussion with a third investigator. We used the Grading of Recommendations, Assessment, Development and Evaluations (GRADE) approach to calculate the certainty of evidence (GRADEpro software, McMaster University and Evidence Prime, Version 2024).

### 2.4. Statistical Methods

Comparisons of mortality and complications in relation to diabetes were presented as risk ratios (RRs) and 95% confidence intervals (CIs) using the Mantel–Haenszel method. Pooled RRs were calculated using a random-effects model. The heterogeneity among the studies was tested using Higgins’ *I*^2^ statistic, where *I*^2^ ≥ 50% indicated heterogeneity. Publication bias was tested using Egger’s test and a funnel plot. A sensitivity analysis was conducted through repeated meta-analyses after excluding each study to determine the robustness of the outcomes. All statistical analyses and graphical presentations were conducted using Comprehensive Meta-Analysis software version 3 (Biostat Inc., Englewood, NJ, USA). Statistical significance was set at *p* < 0.05.

## 3. Results

### 3.1. Study Characteristics

The literature search yielded 2049 studies (PubMed, 784; EMBASE, 1165; Cochrane Library, 100). After excluding 427 duplicate studies and 1583 studies that did not meet the inclusion criteria, 39 studies were included in the meta-analysis (Figure 1). A total of 5,139,360 participants who underwent TKA were enrolled, 21.5% of whom had diabetes. The characteristics of each study are summarized in Table 1.

### 3.2. Risk of Bias Assessment

The results of the risk of bias assessment using RoBANS are summarized in Appendix A. All studies showed a low risk of bias in the selection of participants. Eleven of the thirty-nine studies showed a high risk of bias owing to confounders. All studies demonstrated a low risk of bias concerning the measurement of the intervention, blinding for outcome assessment, incomplete outcome data, and selective outcome reporting. The overall results from the GRADEpro incorporating the assessment of quality of evidence for each outcome are presented in Appendix A.

### 3.3. Diabetes, Periprosthetic Joint Infection, and Prosthesis Revision

Twenty-seven studies were included in the meta-analysis regarding the association between diabetes and periprosthetic joint infection. The RR for periprosthetic joint infection was 1.71 (95% CI: 1.46–2.00, *p* < 0.01; Figure 2A). Significant heterogeneity (*I*^2^ = 84.2%) and publication bias (Egger’s test, *p* = 0.005; Appendix A) were found. In the sensitivity analysis, the significance of the results did not change, even after the removal of each study (Appendix A). A subgroup analysis was performed based on follow-up duration. In eleven studies, the RR for periprosthetic joint infection in the group with a follow-up of 1 year or less is 1.40 (95% CI: 1.18–1.65, *p* < 0.01). In seven studies, the RR for periprosthetic joint infection in the group with a follow-up of more than one year is 2.23 (95% CI: 1.04–5.08, *p* = 0.04). In four studies, the RR for prosthesis revision is 1.37 (95% CI: 1.23–1.52, *p* < 0.01) without heterogeneity (*I*^2^ = 0%; Figure 2B).

### 3.4. Diabetes and Major Adverse Cardiovascular or Cerebrovascular Events

Seven studies were included in the meta-analysis regarding the association between diabetes and CVD. Compared to patients without diabetes, those with diabetes showed a higher risk of CVD after TKA (RR = 2.50, 95% CI: 1.50–4.17, *p* < 0.01) (Figure 3A). Significant heterogeneity was observed among studies (*I*^2^ = 97%). However, the funnel plot was symmetrical and no significant publication bias was detected (Egger’s test, *p* = 0.435) (Appendix A). A sensitivity analysis showed robust results with repeated analysis after excluding each study (Appendix A). In four studies, the RR for CVA was 2.38 (95% CI: 1.48–3.81, *p* < 0.01), with significant heterogeneity (*I*^2^ = 55%) (Figure 3B).

### 3.5. Diabetes and Other Complications

Patients with diabetes showed an elevated risk of systemic infection, encompassing pneumonia (odds ratio [OR]: 1.54, 95% CI: 1.15–2.07, *p* < 0.01 in four studies), UTI (OR: 1.86, 95% CI: 1.07–3.26, *p* = 0.02 in four studies), and sepsis (OR: 1.61, 95% CI: 1.46–1.78, *p* < 0.01 in four studies) (Figure 4A–C). In addition, the presence of diabetes increased the risk of deep vein thrombosis (OR: 1.58, 95% CI: 1.22–2.04, *p* < 0.01), but did not lead to an increased risk of pulmonary embolism (Figure 4D,E).

### 3.6. Diabetes and In-Hospital Death or Death within 3 Months of TKA

Ten studies were included in the meta-analysis of the association between diabetes and mortality. Compared to patients without diabetes, those with diabetes showed a higher risk of in-hospital death or death within 3 months of TKA (RR = 1.28, 95% CI: 1.02–1.60, *p* = 0.04) (Figure 5). No significant publication bias was detected (Egger’s test, *p* = 0.115; Appendix A), and significant heterogeneity was observed among the studies (*I*^2^ = 75.2%). The sensitivity analysis did not show robust results after repeated analyses after excluding each study (Appendix A).

## 4. Discussion

Our meta-analysis showed that (1) patients with DM demonstrated a higher rate of periprosthetic joint infection and prosthesis revision; (2) patients with DM showed an elevated incidence of systemic infections, including pneumonia, UTI, and sepsis; (3) the postoperative risk of CVD/CVA was notably higher in patients with DM than in those without; (4) the presence of DM increased the risk of DVT, but did not increase the risk of PE; and (5) DM was associated with a higher mortality rate within 3 months of TKA. To the best of our knowledge, this is the first study to conduct a large-scale meta-analysis of studies investigating the impact of DM on periprosthetic joint infection and revision rates, systemic infections, DVT/PE, CVD/CVA, and mortality in patients after TKA.

Infection is a leading cause of rehospitalization, revision surgery, and mortality after TKA [12,51,52]. Long-term hyperglycemia exerts adverse effects on the immune system by impairing leukocyte function, thereby increasing the risk of infection [53,54]. Ahmad et al. [55] conducted a meta-analysis and found that patients with DM had a greater risk of superficial and deep site infections after TKA. However, the study only analyzed site infections in 18 studies. The strength of our study lies in the analysis of both surgical sites and systemic infections, which incorporated data from 33 studies. Several studies have indicated that maintaining adequate glucose levels during the perioperative period can reduce postoperative infection rates [56,57]. Our findings support the implementation of perioperative precautions and patient education when performing TKA in patients with DM to mitigate the risk of infection.

Prosthesis revision is associated with poorer functional outcomes than primary TKA and can significantly affect the patient’s quality of life [58]. Aseptic and septic loosening are the two main reasons for prosthesis revision after TKA [59]. DM is known to increase the risk of surgical site infections, which may consequently increase the likelihood of septic loosening [55]. However, Meding et al. [13] demonstrated that patients with DM not only experienced septic loosening, but also had a significantly greater incidence of aseptic loosening after TKA. DM can influence the incidence of aseptic loosening by decreasing bone mineral density due to chronic inflammation [60]. While studies [24,25] suggest an association between the presence of DM and prosthesis revision, this relationship has not been definitively proven by meta-analyses. Our meta-analysis confirmed that DM patients have a significantly higher risk for prosthesis revision than non-DM patients.

CVD/CVA is a fatal complication of TKA. High preoperative glucose levels increase the risk for CVD/CVA by increasing the levels of inflammatory mediators and impairing endothelial cell function [61,62]. Additionally, because of preoperative stress and starvation, patients with DM experience aggravated insulin resistance, known as stress hyperglycemia, which further increases the risk of CVD/CVA [63]. Studies have suggested an association between the presence of DM and the occurrence of CVD/CVA after TKA [15,39]; however, some contradictory studies [12,14] also exist, making this a controversial topic. Our meta-analysis showed that patients with DM had a higher risk of CVD/CVA after TKA than those without DM. Therefore, surgeons should focus on controlling perioperative glucose levels, providing preoperative consultation to reduce patient anxiety, and minimizing starvation to prevent stress hyperglycemia in patients with DM. Furthermore, unusual symptoms such as chest pain, headache, and mental changes should be carefully monitored after TKA in patients with DM.

Our analysis shows that patients with DM have a higher risk of DVT and a similar risk of PE compared to patients without DM following TKA. This finding aligns with previous meta-analyses indicating that patients with DM are more susceptible to DVT after TKA, although this study did not further analyze the effect of DM on PE [16]. DM causes endothelial damage, increases blood coagulation, and impairs fibrinolysis. Elevated levels of fibrinogen, von Willebrand factor, and other endothelium-derived mediators contribute to increased blood viscosity and promote platelet adhesion [64,65]. Considering the significantly increased incidence of DVT in patients with DM, antiplatelet or anticoagulation agents should be prescribed perioperatively. Surgeons should also educate and encourage patients to engage in early ambulation and calf exercises to reduce the risk of postoperative DVT [66,67].

The effect of DM on mortality after TKA remains controversy [12,15,24,37]. In this study, the rate of in-hospital death or death within 3 months after TKA was significantly higher in patients with DM than in those without DM. This finding is consistent with that of previous meta-analyses, which also found a significant difference in in-hospital mortality between DM and non-DM groups [8]. However, this study only analyzed in-hospital mortality and included only two studies, which is insufficient to fully understand the effect of DM on mortality. The low mortality rate after TKA makes it challenging to establish definitive trends in individual studies with small sample sizes. Given that only 6 of the 14 studies in our analysis reported findings related to mortality, meta-analysis offers advantages over individual studies in analyzing low-incidence events such as mortality.

This study had several limitations. First, our findings may vary between patients with uncontrolled DM and those with controlled DM. Most previous studies have not reported HbA1c levels, making it difficult to analyze the effects of DM control [12,15]. However, according to a study by Tarabichi et al. [68], HbA1c levels showed significant differences only in PJI, with no significant differences observed in other complications such as sepsis, thromboembolism, and genitourinary and cardiovascular complications. Further studies that focus on the effects of DM control on various complications are required to clarify this issue. Second, our results may differ between patients with type 1 and type 2 DM. However, most studies have only presented data on type 2 DM, making related analyses difficult [15,25]. Further analysis according to the type of DM may be needed to provide better consultation for patients with specific DM types. Finally, the presence of other confounding factors associated with DM, such as cardiovascular disease, dyslipidemia, and obesity, was not considered. This suggests the need for further research that carefully considers and controls these confounding factors.

## 5. Conclusions

Patients with DM demonstrate a higher rate of postoperative complications and a higher mortality rate within 30 days after TKA. It is crucial to educate patients about the perioperative risks associated with DM and to develop evidence-based guidelines to prevent complications after TKA.

## Figures and Tables

**Figure 1 medicina-60-01757-f001:**
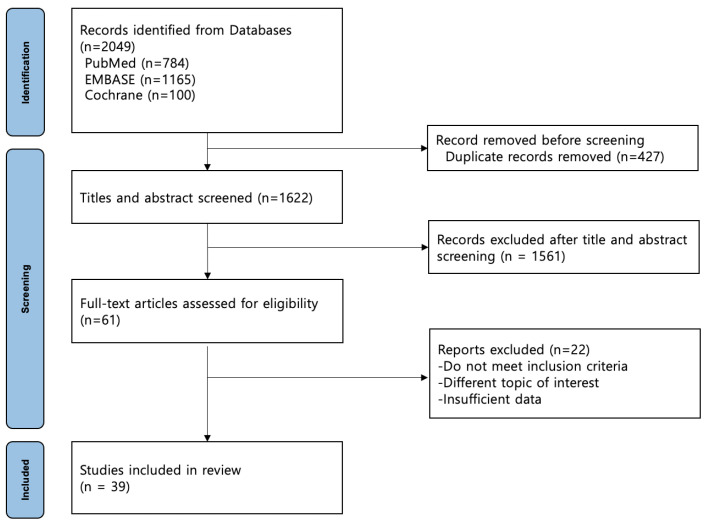
Flow chart of the study population.

**Figure 2 medicina-60-01757-f002:**
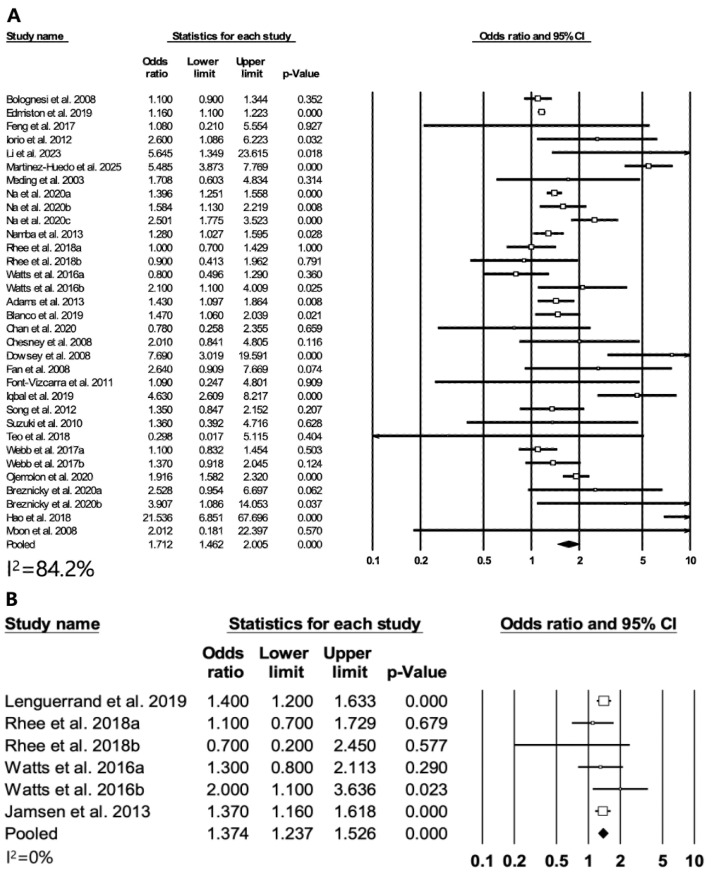
Differences in periprosthetic joint infection and revision rates. Forest plots showing differences between DM patients and non-DM patients. (**A**) Periprosthetic joint infection and (**B**) revision rate [6,9,11,12,13,14,15,20,21,22,23,24,25,26,27,28,29,30,31,32,33,34,35,36,39,45,46,47,49].

**Figure 3 medicina-60-01757-f003:**
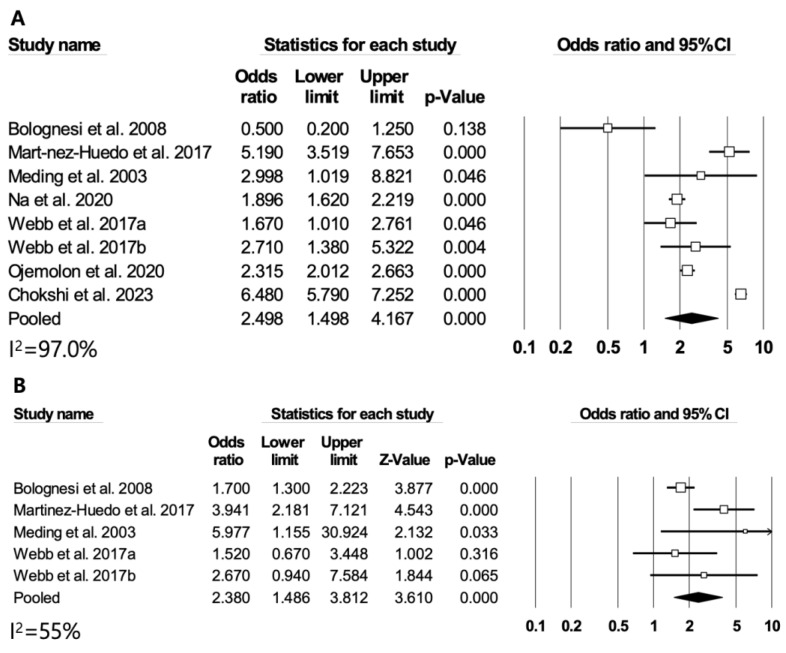
Differences in CVD and CVA between DM patients and non-DM patients. (**A**) CVD and (**B**) CVA [12,13,14,15,23,39,40].

**Figure 4 medicina-60-01757-f004:**
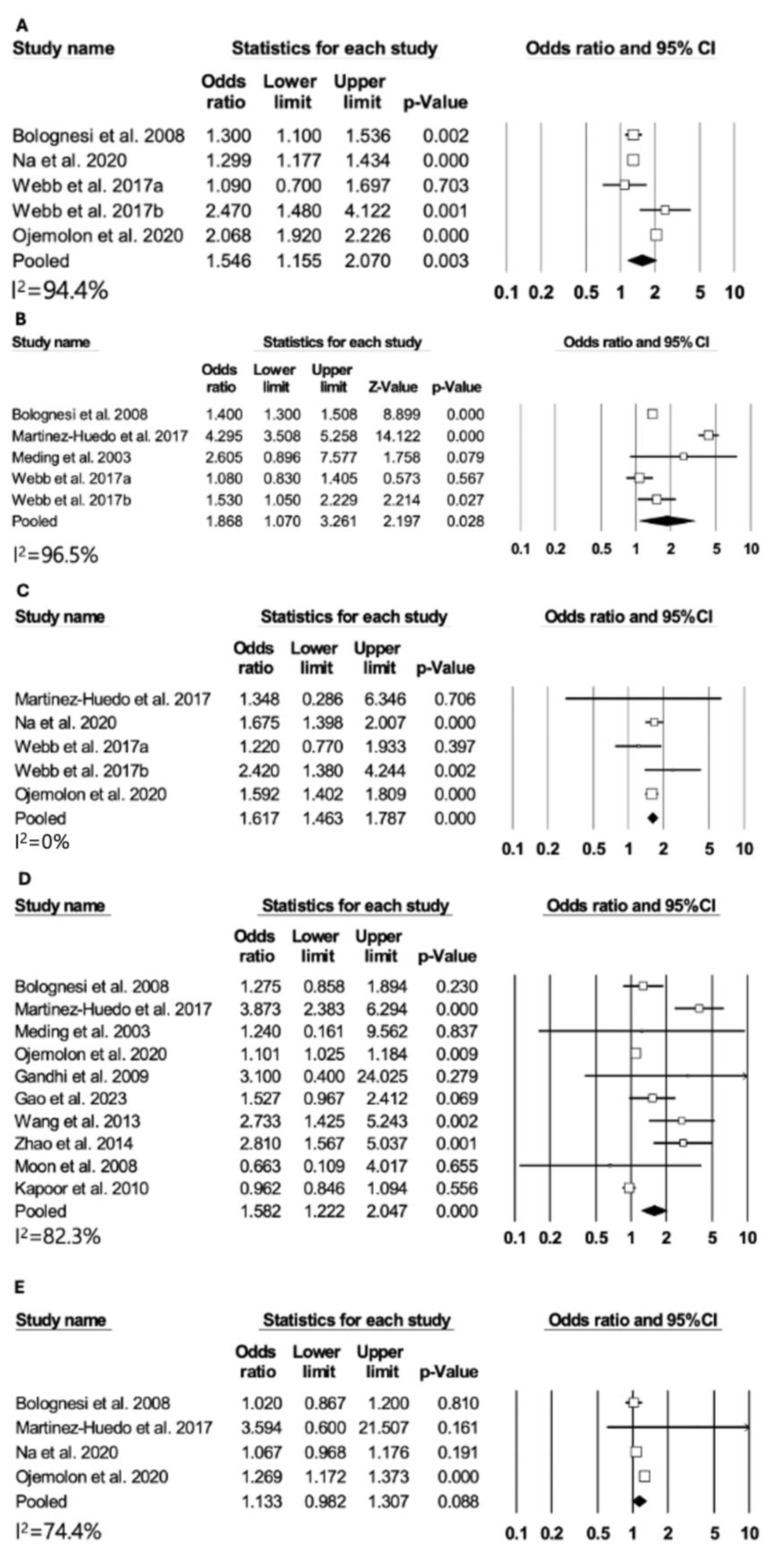
Differences in pneumonia, UTI, sepsis, DVT, and PE between DM patients and non-DM patients. (**A**) Pneumonia, (**B**) UTI, (**C**) sepsis, (**D**) DVT, and (**E**) PE [12,13,14,15,23,39,41,42,43,44,47,50].

**Figure 5 medicina-60-01757-f005:**
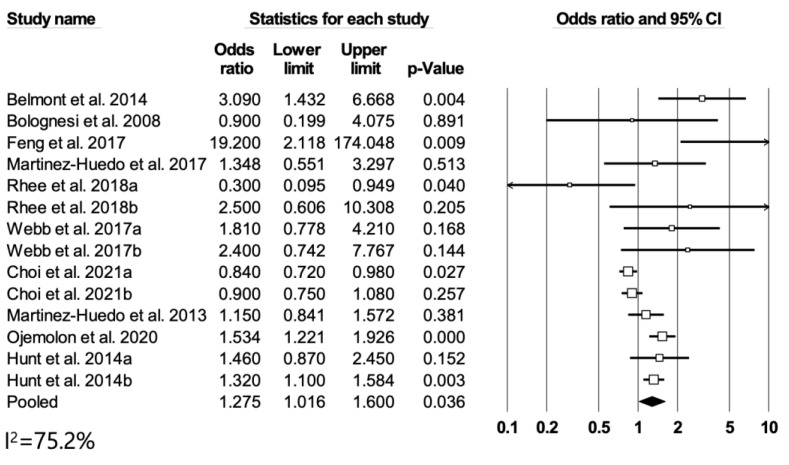
Differences in in-hospital death or death within 3 months. Forest plots showing differences between DM patients and non-DM patients [12,15,19,21,24,37,38,39,48].

**Table 1 medicina-60-01757-t001:** Major characteristics of the included studies.

Study	Year	Country	Types	Sample Size (*n*)	Male (%)	Age (Years)	Number of Patients (DM/non-DM)	Recorded Parameters
Belmont et al. [19]	2014	USA	Cohort	15,321	35.5	67.3	2795/12,526	Mortality
Bolognesi et al. [12]	2008	USA	Cohort	458,943	38.8	67.9	46,044/412,899	Mortality, CVA/CVD, pneumonia, UTI, DVT/PE, infection (deep + superficial)
Edmiston et al. [20]	2019	USA	Cohort	320,648	60.1	64.5	72,789/247,859	Infection (deep + superficial)
Feng et al. [21]	2017	China	Cohort	1542	20.1	65	187/1355	Mortality, infection (deep + superficial)
Iorio et al. [11]	2012	USA	Cohort	2479	N/A	N/A	205/2274	Infection (deep + superficial)
Lenguerrand et al. [22]	2019	UK	Cohort	557,426	43.1	70	66,905/490,521	Revision
Li et al. [6]	2023	China	Case–control	650	N/A	N/A	196/454	Infection (deep + superficial)
Martínez-Huedo et al. [15]	2017	Spain	Cohort	195,355	32.3	71.6	30,571/164,784	Mortality, CVA/CVD, UTI, sepsis, DVT/PE, infection (deep + superficial)
Meding et al. [13]	2003	USA	Cohort	5220	48 in DM 40 in non-DM	70 in DM 70 in non-DM	329/4891	CVA/CVD, UTI, DVT, deep infection
Na et al. [23]	2020	USA	Cohort	355,155	36.4	N/A	81,350/273,805	CVD, pneumonia, UTI, sepsis, PE, infection (deep + superficial)
Namba et al. [9]	2013	USA	Case–control	56,216	37	67.4	14,432/41,784	Deep infection
Rhee et al. [24]	2018	Canada	Cohort	17,243	40.2	67.1	4633/12,610	Mortality, revision, infection (deep + superficial)
Watts et al. [25]	2016	USA	Cohort	1814	35 in DM 29 in non-DM	64 in DM 63 in non-DM	530/1284	Revision, deep infection
Adams et al. [26]	2013	USA	Cohort	40,491	43 in DM 36 in non-DM	68 in DM 68 in non-DM	7567/32,924	Deep infection
Blanco et al. [27]	2019	Spain	Case–control	132	N/A	N/A	37/95	Deep infection
Chan et al. [28]	2020	Hong Kong	Cohort	2017	24	68	898/1119	Deep infection
Chesney et al. [29]	2008	UK	Cohort	1332	47.7	N/A	53/1279	Infection (deep + superficial)
Dowsey et al. [30]	2008	Australia	Cohort	1214	N/A	N/A	206/1008	Deep infection
Fan et al. [31]	2008	Hong Kong	Cohort	472	16.7	69	82/390	Infection (deep + superficial)
Font-Vizcarra et al. [32]	2011	Spain	Case–control	213	23.9	72	36/177	Infection (deep + superficial)
Iqbal et al. [33]	2019	Pakistan	Case–control	4269	34.1	61.4	416/3853	Deep infection
Song et al. [34]	2012	Korea	Case–control	3426	26	67	792/2634	Infection (deep + superficial)
Suzuki et al. [35]	2010	Japan	Case–control	2022	12.5	N/A	276/1746	Deep infection
Teo et al. [36]	2018	Singapore	Case–control	905	21.4	65.9	123/782	Infection (deep + superficial)
Webb et al. [14]	2017	USA	Cohort	114,102	37.3	N/A	20,248/93,854	Mortality, CVA/CVD, pneumonia, UTI, sepsis, DVT/PE, infection (deep + superficial)
Choi et al. [37]	2021	Korea	Cohort	560,954	25.8	N/A	N/A	Mortality
Martínez-Huedo et al. [38]	2013	Spain	Cohort	250,205	26.1	N/A	220,453/29,752	Mortality
Ojemolon et al. [39]	2020	Nigeria	Cohort	1,479,010	38.5	66.5	317,975/1,161,035	Mortality, CVA, pneumonia, sepsis, DVT, infection (deep + superficial)
Chokshi et al. [40]	2023	USA	Cohort	112,531	N/A	N/A	22,767/89,764	CVA
Gandhi et al. [41]	2009	Canada	Cohort	1460	N/A	N/A	N/A	DVT
Gao et al. [42]	2023	China	Case–control	661	33	N/A	91/570	DVT
Wang et al. [43]	2013	China	Cohort	245	32.1 in DM 28.6 in non-DM	67 in DM 67.1 in non-DM	53/192	DVT
Zhao et al. [44]	2014	China	Cohort	358	28.6 in DM 34.7 in non-DM	68.1 in DM 68.2 in non-DM	70/288	DVT
Breznicky et al. [45]	2020	Slovakia	Case–control	150	N/A	N/A	11/139	Infection (deep + superficial)
Hao et al. [46]	2018	China	Cohort	1161	N/A	N/A	152/1009	Infection (deep + superficial)
Moon et al. [47]	2008	Korea	Cohort	342	9.4	67.6	171/171	DVT, infection (deep + superficial)
Hunt et al. [48]	2014	UK	Cohort	363,427	N/A	N/A	41,832/321,595	Mortality
Jamsen et al. [49]	2013	Finland	Cohort	53,007	29	70.3	3965/49,042	Revision
Kapoor et al. [50]	2010	USA	Case–control	158,140	35.1	N/A	26,458/131,682	DVT

DM, diabetic mellitus; N/A, not applicable; USA, United States of America; CVA, cerebrovascular accident; CVD, cardiovascular disease; UTI, urinary tract infection; DVT, deep vein thrombosis; PE, pulmonary embolism; UK, United Kingdom.

## Data Availability

All data from this study are available upon reasonable request to the corresponding author.

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
