# Peer review of "Influence of Diabetes Mellitus on Postoperative Complications After Total Knee Arthroplasty: A Systematic Review and Meta-Analysis"

_medicina, 2024, doi:10.3390/medicina60111757_

Round 1
Reviewer 1 Report
Comments and Suggestions for Authors
The manuscript titled "Influence of Diabetes Mellitus on Postoperative Complications After Total Knee Arthroplasty: A Systematic Review and Meta-Analysis" presents a review of the literature on the topic of diabetes as a potential enemy of the outcomes of patients with prosthesis implantation. There are some issues in the work that should be clarified/modified:
- The introduction is too superficial and does not adequately focus on the core of the work. I suggest revising it.
- The quality of the table should be improved both in terms of the information contained and in terms of formatting so that they are more easily readable and more aesthetically pleasing.
- Why did you choose to always use a random effects model and not a fixed effects model in the meta-analysis?
- In the statistical analysis chapter, the p-value threshold considered for the significance of the results should be indicated.
- The literature search dates back to almost a year ago. To increase the significance of the study and interest in a work with updated literature, I suggest conducting a new literature search to identify any significant new studies published in the last year.
Reviewer 2 Report
Comments and Suggestions for Authors
Dear authors
This is a well conducted review. However I would like you to interpret the results based on the level of evidence you provide. In other words uncertainty should be introduced if the quality of evidence is not level 1. To find out you will need to assess it with the appropriate tools such as Grade or cinema. Also you need to mention measure of variability in the abstract such as confidence intervals. I would be happy to review again once you have done those major changes.
Comments on the Quality of English LanguageNone
Reviewer 3 Report
Comments and Suggestions for Authors
the study is very well conducted and written
-please be sure that references are correctly formatted and up to date
-line 56: report the hypothesis
-I suggest the author to analyze more in dept the time frame between suregy and all complications if possible
Comments on the Quality of English Language
good
Round 2
Reviewer 1 Report
Comments and Suggestions for Authors
The work has been appropriately modified and can be accepted for the publication.